# Differential Diagnosis of Intraplaque Hemorrhage and Dissection on High-Resolution MR Imaging in Patients with Focal High Signal of the Vertebrobasilar Artery on TOF Imaging

**DOI:** 10.3390/diagnostics11061024

**Published:** 2021-06-03

**Authors:** Jung Hwan Kim, Hyo Sung Kwak, Seung Bae Hwang, Gyung Ho Chung

**Affiliations:** Department of Radiology and Research Institute of Clinical Medicine of Jeonbuk National University, Biomedical Research Institute of Jeonbuk National University Hospital, Jeonju 54896, Korea; hanbobo@hanmail.net (J.H.K.); sbh1010@jbnu.ac.kr (S.B.H.); chunggh@jbnu.ac.kr (G.H.C.)

**Keywords:** intracranial artery, atherosclerosis, MRI, vessel wall imaging, dissection

## Abstract

Purpose: Intraplaque hemorrhage (IPH) and dissection in the vertebrobasilar artery (VBA) on time of flight (TOF) source imaging are seen as focal eccentric high-signal intensity. The purpose of this study is to identify IPH and dissection in the VBA using high-resolution magnetic resonance imaging (HR-MRI). Methods: A total of 78 patients (VBA IPH: 55; dissection: 23) with focal high-signal intensity in the VBA on simultaneous non-contrast angiography and intraplaque hemorrhage (SNAP) of HR-MRI were included in this study. The focal high-signal intensity in the VBA on SNAP was defined as >200% than that of the adjacent muscle. We analyzed the signal intensity ratio (area of focal high signal intensity area/lumen) on TOF imaging and black blood (BB) T2-weighted imaging. Results: The VBA IPH group was older than the dissection group and had more hypertension. Signal intensity of a false lumen in patients with dissection on TOF imaging was significantly higher than that of VBA IPH (*p* < 0.001). The signal intensity ratio between lumen and lesion on TOF imaging was significantly higher in the dissection group (*p* < 0.001). The signal intensity of a false lumen in patients with dissection on BB T2-weighted imaging was significantly lower than that of VBA IPH (*p* < 0.001). The signal intensity ratio between lumen and lesion on BB T2–weighted imaging was significantly higher in the VBA IPH group (*p* < 0.001). Conclusions: TOF imaging and BB T2-weighted imaging on HR-MRI in patients with focal eccentric high-signal intensity on TOF imaging can distinguish between VBA IPH and dissection.

## 1. Introduction

The posterior cerebral circulation, also known as the vertebrobasilar system, consists of the basilar and vertebral arteries and contributes about 20% of the brain’s blood supply [1]. Posterior circulation ischemic stroke accounts for only about 20–30% of all ischemic stroke, but it has higher mortality and morbidity rates compared with anterior circulation ischemic stroke [2,3]. It is known that the major cause of vertebrobasilar artery (VBA) stenosis is atherosclerosis with or without intraplaque hemorrhage (IPH) [4]. However, vertebrobasilar artery dissection accounts for some VBA stenosis.

IPH is thought to be caused by the rupture of fragile and tortuous neovessels formed within the plaque [5]. IPH is considered to be an important atherogenic stimulus that advances macrophage infiltration, which makes the plaque more unstable [6,7]. Dissection is a rare cause of stroke; however, it represents one of the more common causes of stroke in patients younger than 45 years of age [8,9]. Especially, it frequently occurs in the carotid artery or vertebral artery with high mobility. Dissection was not easy to diagnose in the past, but the diagnosis is becoming easier as vascular imaging technology has developed [10,11,12].

IPH has a different etiology from dissection, and it requires different treatment strategies to treat patients with suspected VBA stenosis. However, in general, atherosclerotic plaque or dissection is observed as simple stenosis in time-of-flight (TOF) MR angiography (MRA) [13,14]. On the other hand, high-resolution magnetic resonance imaging (HR-MRI) is very useful for showing various blood vessel conditions. Therefore, it is expected that changes caused by various etiologies, such as atherosclerosis or dissection, can be distinguished on HR-MRI [15]. IPH and dissection in the VBA on TOF imaging is seen as focal eccentric high-signal intensity [15]. The differences in signal intensity between IPH and dissection in the VBA on HR-VWI are not fully known. The aim of this study was to distinguish between VBA IPH and dissection by signal intensity on TOF imaging and black blood (BB) T2-weighted imaging on HR-VWI.

## 2. Materials and Methods

### 2.1. Patients

This study was approved by the local institutional review board, and informed consent was obtained from all patients before imaging. Between January 2015 and December 2019, we consecutively selected patients for VBA stenosis related to atherosclerosis or suspicious dissection using TOF-MRA. During this period, all patients underwent standard brain MRI and MRA to detect any other lesions. We performed HR-MRI for evaluation of VBA lesions within 1 week after the initial MR examination. Patients with any of the following features were excluded from the analysis: (1) Massive and dominant calcified plaque; (2) complete BA occlusion on MRA and HR-MRI; (3) other vasculopathies, such as inflammatory arteritis or Moyamoya disease; (4) dissecting aneurysm; (5) clinical contraindications to MRI; (6) no IPH in the VBA atherosclerosis; or (7) insufficient MR imaging quality to evaluate contrast enhancement of the aneurysmal wall.

### 2.2. High-Resolution MR Imaging

MRI was performed with a 3T MRI scanner (Achieva; Philips Medical Systems, Amsterdam, The Netherlands) with a 16-channel head coil. All patients initially underwent conventional brain MRI, which included 3D TOF-MRA. TOF-MRA of the axial plane was obtained for each patient, and data were reconstructed using a dedicated online postprocessing tool to determine blood vessel architecture.

The HR-MRI protocol included 5 MR scans: BB T1-weighted, BB T2-weighted, TOF axial, simultaneous non-contrast angiography and intraplaque hemorrhage (SNAP), and contrast-enhanced BB T1-weighted imaging. BB T1-weighted imaging was acquired using a 2D turbo spin-echo sequence with the following imaging parameters: repetition time/echo time = 800/10 ms, field of view = 140 × 140 mm, matrix size = 140 × 150, slice thickness = 2.0 mm, echo train length = 10, and number of excitations = 2. Gadodiamide (0.1 mmol/kg body weight; Dotarem; Guerbet, Aulnay-sous-Bois, France) for contrast-enhanced BB T1-weighted imaging was injected as a bolus intravenously in all patients. Contrast-enhanced BB T1-weighted imaging was carried out ~5 min after contrast injection. SNAP sequence was performed for evaluation of optimal IPH as hyperintense. Image parameters were as follows: TR/TE/TI = 10/4.7/490 ms, FA = 11°, ETL = 98, FOV = 149 × 149 mm, matrix = 187 × 216. The total scan time was 25 to 30 min, and patients remained in the MR machine for 35 to 45 min.

### 2.3. Clinical Data Assessment

Patients were classified as having either a symptomatic or asymptomatic lesion according to the presence of recent ischemic stroke. Symptomatic stenosis was defined as a diffusion-restrictive lesion seen on DWI in the territory of the stenotic VBA with a corresponding acute neurologic deficit within 2 weeks before MR imaging. Clinical data, including basic demographics and risk factors for atherosclerosis, namely diabetes, hypertension, dyslipidemia, current smoking, and history of coronary disease, were also recorded.

### 2.4. Image Analysis

All MR images were reviewed retrospectively by 2 neuroradiologists (with 25 years and 15 years of experience, respectively) blinded to the clinical information of each patient. They assessed image quality by consensus using a 4-scale scoring system (1, poor; 2, adequate; 3, good; 4, excellent). Images with a score of 1 were excluded from the final analysis. Disagreements regarding image quality were resolved by consensus.

VBA Plaque was defined as a thickening of the focal wall relative to image slices from beneath or above the focal wall, as identified on BB T2- and T1-weighted imaging. VBA IPH was defined as a signal intensity greater than 150% of T1 SNAP signal of adjacent muscle [16]. VBA dissection was defined as a wall thickening with low-signal intensity because of an intimal flap on BB T2-weighted imaging and high-signal intensity because of an intimal flap on TOF-MRA [17,18]. We analyzed the presence of VBA IPH and dissection among all samples using HR-MRI. Two neuroradiologists performed all these procedures. Consensus interpretation was used for the final analysis when the readers’ interpretations differed.

The percentage of stenosis was estimated on TOF-MRA using the formula of [(a − b)/a] × 100% (a = narrowed vessel diameter, b = proximal normal vessel diameter). Because of the small diameters of intracranial vessels, we focused on stenoses involving the larger, more proximal intracranial arteries. In addition, maximal wall thickness was measured at the highest point of plaque on T1-weighted imaging.

We measured the signal intensity of the VBA plaque and vessel lumen on TOF source imaging and BB T2-weighted imaging. The signal intensity ratio between VBA plaques and vessel lumen on BB T2-weighted imaging and TOF source imaging was measured at the stenotic area.

### 2.5. Statistical Analysis

The proportion of patients with VBA plaque with VBA IPH and dissection was investigated. The demographics and risk factors were compared to these patients. Pearson’s chi-square test and Student’s *t-*test were used for categorical and continuous variables as appropriate. A 2-sided *p-*value of <0.05 was considered statistically significant. Receiver-operating characteristic (ROC) curves of IPH or dissection were assessed to evaluate the diagnostic efficacy for detecting VBA IPH on BB T2-weighted imaging and TOF source imaging. Statistical analysis was performed by using SPSS 23.0 for Windows (SPSS, IBM, Chicago, IL, USA) and MedCalc software, version 16.4.2 (MedCalc, Ostend, Belgium).

## 3. Results

### 3.1. Patients

During the study period, 155 patients underwent HR-MRI for evaluation of VBA plaques or stenosis. Of these patients, 58 were excluded from this study due to no IPH in the VBA atherosclerosis, 7 because of poor quality, 6 because of dissecting aneurysms, and 7 because of massive calcification in the VBA plaques. In total, 78 patients (45 males; mean age, 67.9 years) with VBA IPH or dissection were enrolled. Of these patients, 55 had VBA IPH and 23 had VBA dissection.

### 3.2. Comparison of Clinical Characteristics between IPH and Dissection Groups

The results of the comparison between patients with VBA IPH and dissection are shown in Table 1**.** The mean age of patients with VBA dissection was significantly younger, while IPH patients with VBA plaque were the oldest (56.6 ± 3.1 years old for VBA dissection, 72.7 ± 1.2 years old for VBA plaque with IPH; *p* < 0.001). When the risk factors of atherosclerosis were compared, VBA plaque with IPH patients had the highest levels of hypertension (85.5% vs. 47.8%; *p* < 0.001). Previous stroke history was the most prevalent in IPH groups (25.2% vs. 4.3%; *p* = 0.055). Symptomatic lesions were similar between the two groups.

### 3.3. Signal Intensity Ratio of BB T2-Weighted and TOF Imaging between IPH and Dissection Groups

The single intensity ratios of VBA plaques on BB T2-weighted and TOF imaging between IPH and dissection groups are shown in Table 2. VBA dissection groups on TOF imaging had significantly brighter signal intensity compared to VBA IPH groups (608.4 ± 64.6 vs. 326.6 ± 24.4; *p* < 0.001). In addition, the lumen-lesion signal intensity ratio on TOF imaging was significantly higher in VBA dissection groups (80.7 ± 6.8 vs. 45.4 ± 2.4; *p* < 0.001). VBA IPH groups on BB T2-weighted imaging had significantly bright signal intensity compared to VBA dissection groups (540.9 ± 53.5 vs. 319.8 ± 124.0; *p* < 0.001). In addition, the lesion-lumen signal intensity ratio on BB T2-weighted imaging was significantly higher in VBA IPH groups (1095.5 ± 162.0 vs. 285.2 ± 81.6; *p* < 0.001). The maximal wall thickness and degree of stenosis were similar between the two groups.

### 3.4. Comparison of ROC Analysis between BB T2-Weighted and TOF Imaging

The ROC curve analysis of the signal intensity ratio between both groups is shown in Table 3. The ROC curve analysis revealed a higher diagnostic value for analysis between VBA IPH and dissection for both imaging modalities (TOF imaging: AUC = 0.852, 95% CI = 0.760–0.943, *p* < 0.001; BB T2-weighted imaging: AUC = 0.851, 95% CI = 0.743–0.958, *p* < 0.001) (Figure 1 and Figure 2).

## 4. Discussion

In this study, VBA IPH showed a significantly lower signal intensity compared to a false lumen of dissection on TOF source imaging, and a false lumen of dissection showed a significantly lower signal intensity compared to VBA IPH on BB T2-weighted source imaging. VBA IPH and dissection can be differentiated on BB T2-weighted and TOF source imaging on HR-VWI.

Atherosclerotic plaque is the most important cause of stroke, and is commonly associated with unstable plaque. Especially, IPH is a common feature of atherosclerotic plaques and is an important feature of complex lesions preceding acute stroke events [19]. IPH presence plays a more important role in the progression of plaque during the long term [20,21] and increases the risk of ipsilateral stroke events [16,22]. Therefore, early detection of atherosclerotic IPH is very useful for the management of atherosclerosis and the prevention of stroke.

Patients with intracranial dissection most often present with a nonspecific headache followed by ischemic stroke or subarachnoid hemorrhage (SAH). Dissecting aneurysms related to SAH and transmural dissection show a fusiform or irregular dilatation by extension toward the adventitia [23]. Sub-intimal dissection related to ischemic stroke shows a focal stenosis/occlusion or strings of pearls sign [23]. Thus, identifying dissecting aneurysms on imaging studies is easier for correct diagnosis and management. However, dissection presented by focal stenosis or complete occlusion is very difficult to distinguish between other stroke causes. In this study, as mentioned above, patients with a dissecting aneurysm were excluded from the analysis. In addition, in our study, young age was a factor significantly affecting VBA dissection. However, hypertension was a factor that significantly affected VBA IPH.

Detection of IPH commonly uses T1-weighted MR imaging techniques such as fast spin-echo imaging, TOF, MPRAGE imaging, or SNAP imagings. Ota et al. [24] reported that among T1-weighted images, MPRAGE sequences demonstrated superior diagnostic accuracy for detecting and quantifying carotid IPH compared to other T1 weighted imaging modalities and compared to histology. According to Li et al. [25,26], SNAP could detect carotid IPH with a smaller size and has higher sensitivity in detecting hemorrhage compared with MPRAGE in a clinical and histological study. We used SNAP imaging as a reference for the detection of VBA IPH.

The radiological features of arterial dissection are the intimal flap, double lumen, intramural hematoma, pseudoaneurysm, stenosis, and occlusion. However, because these were formed primarily by using luminal angiography, these criteria may not apply to HR-MRI [27]. In a study by Wang et al., intramural hematoma (61% of patients), double lumen (50% of patients), and intimal flap (42% of patients) were recognized as the characteristic features of dissection by high-resolution MRI [28]. However, recent studies in patients with middle cerebral artery dissection reported that features of a double lumen and intimal flap are most common on HR-MRI [18,29,30,31]. When HR-MRI shows intramural hematoma without a double lumen or intimal flap, intracranial artery dissection should be considered first in the differential diagnosis. However, an interpreter may have to take into account IPH in equivocal cases that also demonstrate the clinical features of atherosclerosis. The differentiation between IPH and intramural hematoma may sometimes be difficult and may rely on clinical information such as patient age, atherosclerotic risk factors, symptoms, and change of radiologic findings on follow-up [15].

In this study, we investigated the signal intensity of IPH and dissection in patients with VBA stenosis on BB T2-weighted imaging and TOF imaging. The two categories sometimes showed vascular hyperintensity on T1-weighted imaging because of an IPH or intramural hematoma [32]. In addition, TOF source imaging showed a high signal intensity in an IPH or intramural hematoma. In our study, the signal intensity of a false lumen in the dissection group on TOF imaging showed a bright high signal intensity compared to the signal intensity of IPH. A false lumen of dissection on BB T2-weighted imaging showed a dark signal intensity by flow suppression of the BB technique compared to the signal intensity of plaque. These findings suggest that a false lumen of dissection might result in continuous or turbulent flow rather than a clot or thrombus due to flow suppression from the BB technique and similar signal intensity of a true lumen on TOF imaging and BB T2-weighted imaging may be seen. The diagnostic performance of detection of VBA dissection using our criteria revealed a higher value for both imaging modalities in ROC analysis.

There are some limitations to this study. First, the 78 patients with 55 VBA IPH and 23 VBA dissection were retrospectively recruited. Future studies with larger samples are needed. Second, this retrospective study has a nonrandomized study design; the possibility exists that patients in this study may constitute a high-risk subgroup who require MR images of the head and neck. Finally, a final diagnosis of IPH and dissection was performed by trained neuroradiologists using HR-MRI without histological analysis. The diagnosis of intracranial vessel disease based on MR imaging without histological analysis is a problem that cannot be solved.

## 5. Conclusions

VBA IPH shows a relatively low signal intensity compared to a false lumen of dissection on TOF imaging, and a false lumen of dissection showed a relatively low signal intensity compared to VBA IPH on BB T2-weighted imaging on HR-MRI. TOF imaging and BB T2-weighted imaging on HR-MRI in patients with focal eccentric high-signal intensity on TOF imaging can distinguish between VBA IPH and dissection.

## Figures and Tables

**Figure 1 diagnostics-11-01024-f001:**
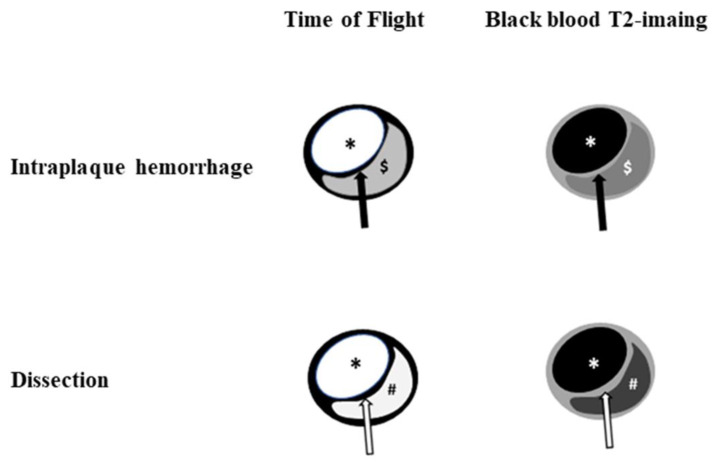
Illustration of HR-VWI between intraplaque hemorrhage and dissection. Dissection shows a similar signal intensity between the true lumen and false lumen on time of flight and black blood T2-weighted imaging. Note: black arrow, fibrous cap; white arrow, intimal flap; *, lumen, $**,** intraplaque hemorrhage; #, false lumen.

**Figure 2 diagnostics-11-01024-f002:**
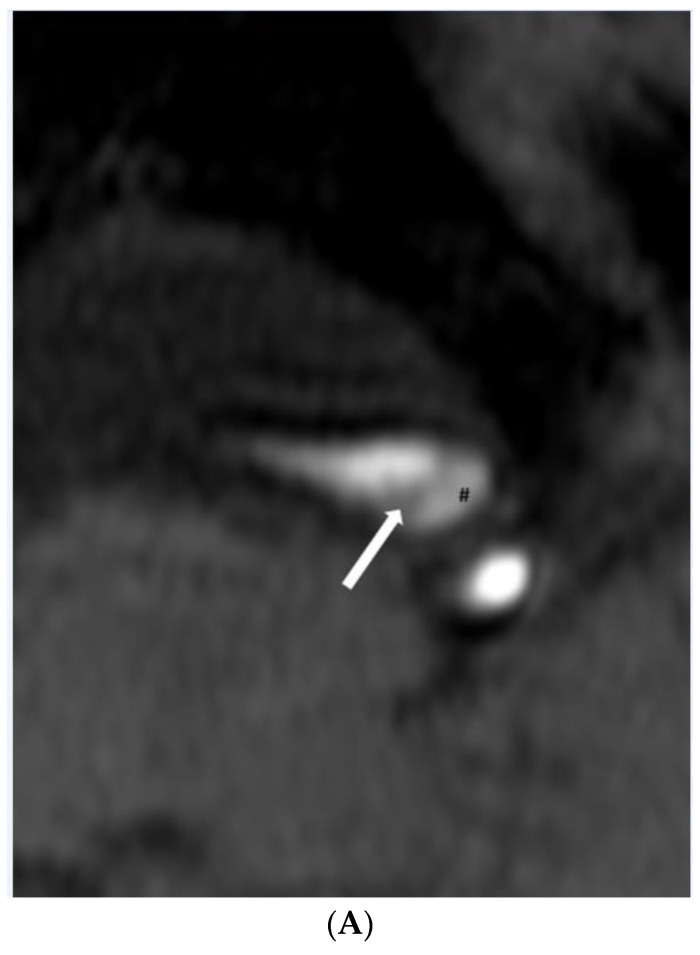
(**A**,**B**). Dissection. (**A**) Time-of-flight imaging shows a similar high signal intensity between the true lumen and false lumen (#). Note the intimal flap with low signal intensity (white arrow). (**B**) Black blood T2-weighted imaging shows a similar low signal intensity of true and false lumen (#) between the intimal flap (white arrow). (**C**,**D**). Intraplaque hemorrhage. (**C**) Intraplaque hemorrhage ($) on time-of-flight imaging shows a relatively subtle high signal intensity compared to the high signal intensity of the lumen. Note the fibrous cap with low signal intensity (white arrow). (**D**) Plaque of basilar artery on black blood T2-weighted imaging shows the iso signal intensity compared to the low signal intensity of the lumen.

**Table 1 diagnostics-11-01024-t001:** Baseline characteristics between VBA IPH and VBA dissection.

	All Patients (*n* = 78)	VBA Atherosclerotic Plaque (*n* = 55)	VBA Dissection (*n* = 23)	*p-*Value
Age (years)	67.9 ± 1.5	72.7 ± 1.2	56.6 ± 3.1	<0.001
Male, n (%)	45 (57.7)	31 (56.4)	14 (60.9)	0.713
Hypertension, n (%)	58 (74.4)	47 (85.5)	11(47.8)	<0.001
Diabetes mellitus, n (%)	23 (29.5)	18 (32.7)	5 (21.7)	0.332
Hyperlipidemia, n (%)	25 (32.1)	19 (34.5)	6 (26.1)	0.465
Alcohol drinking, n (%)	26 (33.3)	17 (30.9)	9 (39.1)	0.482
Smoking, n (%)	23 (29.5)	14 (25.5)	9 (39.1)	0.227
Previous stroke, n (%)	15 (19.2)	14 (25.5)	1 (4.3)	0.055
Previous heart disease, n (%)	17 (21.8)	14 (25.5)	3 (13.0)	0.229
Symptomatic, n (%)	32 (41.0)	21 (38.2)	11 (47.8)	0.430

VBA, vertebrobasilar artery; IPH, intraplaque hemorrhage.

**Table 2 diagnostics-11-01024-t002:** High-resolution MR imaging between VBA IPH and VBA dissection.

	VBA Atherosclerotic Plaque (*n* = 55)	VBA Dissection (*n* = 23)	*p-*Value
Maximal wall thickness, mm	2.0 ± 1.7	2.0 ± 1.1	0.817
Stenosis, %	48.5 ± 2.28	52.8 ± 3.2	0.297
Lumen SI on TOF	753.1 ± 53.8	750.9 ± 63.2	0.981
Lesion SI on TOF	326.6 ± 24.4	608.4 ± 64.6	<0.001
SI ratio on TOF	45.4 ± 2.4	80.7 ± 6.8	<0.001
Lumen SI on BB T2 imaging	113.0 ± 15.8	95.7 ± 29.3	0.525
Lesion SI on BB T2 imaging	540.9 ± 53.5	319.8 ± 124.0	<0.001
SI ratio on BB T2 imaging	1095.5 ± 162.0	285.2 ± 81.6	<0.001
Wall enhancement, n (%)	5 (5.5)	3 (13.0)	0.353

VBA, vertebrobasilar artery; IPH, intraplaque hemorrhage; TOF, time of flight; BB, black blood; SI, signal intensity.

**Table 3 diagnostics-11-01024-t003:** ROC curve analysis of signal intensity ratio between both groups.

	AUC	95%CI	*p-*Value	Sensitivity	Specificity	PPV	NPV	Cutoff
TOF: SI ratio: Lumen/Plaque	0.852	0.760–0.943	<0.001	0.739	0.945	1	0.775	217.4
T2: SI ratio: Plaque/Lumen	0.851	0.743–0.958	<0.001	0.826	0.727	0.333	0.917	53.4

AUC: area under curve, PPV: positive predictive value, NPV: negative predictive value.

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
