# Peer review of "Differential Diagnosis of Intraplaque Hemorrhage and Dissection on High-Resolution MR Imaging in Patients with Focal High Signal of the Vertebrobasilar Artery on TOF Imaging"

_diagnostics, 2021, doi:10.3390/diagnostics11061024_

Round 1

Reviewer 1 Report

This is a clincal investigation to assess the risk of dissection as a rare cause of stroke. While the study is well written, this is limited to a clinical analysis of data normally collected during MRI with limited novelty. The following minor comments needs to be addressed.

page 1 sentence ending with [10,11]: Using medical imanging including CT ans MRA, the strain of the vessel can be determined and this parameter has been correlated with aortic dissection. Please cite the following paper. 

Pasta S, Agnese V, Di Giuseppe M, Gentile G, Raffa GM, Bellavia D, Pilato M. In Vivo Strain Analysis of Dilated Ascending Thoracic Aorta by ECG-Gated CT Angiographic Imaging. Ann Biomed Eng. 2017 Dec;45(12):2911-2920. doi: 10.1007/s10439-017-1915-4. Epub 2017 Sep 7. PMID: 28884233.

Page 2 section 2.2: Please consider to reduce the text reporting all parameter and setting of the MRI study. This appears to long. 

Author Response

Rev 1-1) page 1 sentence ending with [10,11]: Using medical imanging including CT ans MRA, the strain of the vessel can be determined and this parameter has been correlated with aortic dissection. Please cite the following paper. 

Answer) We added a reference about your comment.

Page 2 section 2.2: Please consider to reduce the text reporting all parameter and setting of the MRI study. This appears to long. 

Answer) We deleted T2 and TOF MR parameters because of routine protocol. However, we included the SNAP parameter because of relatively new protocil

Reviewer 2 Report

The paper written by the following Authors: Jung Hwan Kim, Hyo Sung Kwak, Seung Bae Hwang and Gyung Ho Chung, entitled “Differential diagnosis of intraplaque hemorrhage and dissection on high-resolution MR imaging based on focal high signal intensity in the vertebrobasilar artery on TOF MR angiography” presents an interesting study dedicated to application of high-resolution magnetic resonance imaging for Intraplaque hemorrhage and dissection in the vertebrobasilar artery identification.

Although the paper is interesting, I have some major concerns:

Title

The title reflects the results presented here.

Abstract

The abstract is lacking the informative conclusion. It should be written in more details.

Introduction

Authors should add a paragraph on image processing For instance the following citation may be considered:

Shape and Enhancement Analysis as a Useful Tool for the Presentation of Blood Hemodynamic Properties in the Area of Aortic Dissection

  1. Clin. Med.2020, 9(5), 1330; https://doi.org/10.3390/jcm9051330

Conclusions

Conclusions should be extended and related to the results.

Figure 1 and Figure 2

Marks placed on the figures are not visible. The size should be increased. Perhaps authors should consider application of colour marks.

Author Response

2-1) Title: The title reflects the results presented here.

    Answer) Our paper had the purpose of differentiation between dissection and IPH using HR-MRI in patients with focal high signal of the VBA on TOF imaging. Therefore, we had a lot of trouble inferring the title as a result. So, we changed the title as follows.

“Differential diagnosis of intraplaque hemorrhage and dissection on high-resolution MR imaging in patients with focal high signal of the vertebrobasilar artery on TOF imaging”

2-2)Abstract: The abstract is lacking the informative conclusion. It should be written in more details.

Answer) We made soma change of conclusion.

        I agree about your comment, but main results of our paper were described into the results. So, same sentences can be described in the results and conclusion. Therefore, we make some changes.

2-3) Introduction: Authors should add a paragraph on image processing For instance the following citation may be considered: Shape and Enhancement Analysis as a Useful Tool for the Presentation of Blood Hemodynamic Properties in the Area of Aortic Dissection Clin. Med.2020, 9(5),   1330; https://doi.org/10.3390/jcm9051330

  Answer) I knew this paper. However, this paper used the CT algorithm for evaluation of aortic dissection. Our paper focused the differentiation between dissection and IPH using HR-MRI. I thought that it was different the methodology for dissection analysis between CT and MR imaging.

2-4) Conclusions: Conclusions should be extended and related to the results.

 Answer) We made soma change of conclusion.

2-5) Figure 1 and Figure 2: Marks placed on the figures are not visible. The size should be increased. Perhaps authors should consider application of colour marks.

  Answer) We made the color figures for your comment, but color mark is lower contrast compared to black and white marks. Also, change of the mark size did not show the false lumen or IPH. So, we change the figure 1 because of ill defined visualization of words.

Reviewer 3 Report

This is a well written manuscript. Prior suggestions for changes has been met.

Author Response

Thanks

Round 2

Reviewer 2 Report

I accept the present form of the manuscript.